# Sampling alternative conformational states of transporters and receptors with AlphaFold2

**Diego del Alamo[1,2†], Davide Sala[3†], Hassane S Mchaourab[1]\*, Jens Meiler[2,3]\***

[1]Department of Molecular Physiology and Biophysics, Vanderbilt University, Nashville, United States; [2]Department of Chemistry, Vanderbilt University, Nashville, United States; [3]Institute for Drug Discovery, Leipzig University, Leipzig, Germany

**Abstract** Equilibrium fluctuations and triggered conformational changes often underlie the functional cycles of membrane proteins. For example, transporters mediate the passage of molecules across cell membranes by alternating between inward- and outward-facing states, while receptors undergo intracellular structural rearrangements that initiate signaling cascades. Although the conformational plasticity of these proteins has historically posed a challenge for traditional *de novo* protein structure prediction pipelines, the recent success of AlphaFold2 (AF2) in CASP14 culminated in the modeling of a transporter in multiple conformations to high accuracy. Given that AF2 was designed to predict static structures of proteins, it remains unclear if this result represents an underexplored capability to accurately predict multiple conformations and/or structural heterogeneity. Here, we present an approach to drive AF2 to sample alternative conformations of topologically diverse transporters and G-protein-coupled receptors that are absent from the AF2 training set. Whereas models of most proteins generated using the default AF2 pipeline are conformationally homogeneous and nearly identical to one another, reducing the depth of the input multiple sequence alignments by stochastic subsampling led to the generation of accurate models in multiple conformations. In our benchmark, these conformations spanned the range between two experimental structures of interest, with models at the extremes of these conformational distributions observed to be among the most accurate (average template modeling score of 0.94). These results suggest a straightforward approach to identifying native-like alternative states, while also highlighting the need for the next generation of deep learning algorithms to be designed to predict ensembles of biophysically relevant states.

**\*For correspondence:**
hassane.mchaourab@vanderbilt.edu (HSM);
jens.meiler@vanderbilt.edu (JM)

†These authors contributed equally to this work

**Competing interest:** The authors declare that no competing interests exist.

## Editor's evaluation

del Alamo and colleagues illustrate that restricting the depth of the input multiple sequence alignment allows AlphaFold2 to predict diverse conformational ensembles of transporters and receptors, as opposed to single static models reflecting individual states. Although they are limited to a small number of test cases of membrane proteins, the examples are of interest to members of the community. This work presents a validation of a simple approach that may be applicable to all proteins and is thus an exciting advance that is expected to be of broad interest.

## Introduction

Dynamic interconversion between multiple conformations underpins the functions of integral membrane proteins in all domains of life (*Campbell et al., 2016*; *Cournia et al., 2015*; *Shaw et al., 2010*; *Boehr et al., 2009*). For example, the vectorial translocation of substrates by transporters is

mediated by movements that open and close extra- and intracellular gates (*Drew and Boudker, 2016*; *Kazmier et al., 2017*). For G-protein-coupled receptors (GPCRs), ligand binding on the extracellular side triggers structural rearrangements on the intracellular side that initiate downstream signaling (*Wang et al., 2020*; *Gusach et al., 2020*). Traditional computational prediction pipelines reliant on inter-residue distance restraints calculated from deep multiple sequence alignments (MSAs) have historically struggled to accurately predict the structures of these proteins and their movements. The resulting models are unnaturally compact and frequently distorted, preventing critical questions about ligand and/or drug binding modes from being addressed (*Ovchinnikov et al., 2015*; *Nicoludis and Gaudet, 2018*).

A performance breakthrough was unveiled during CASP14 by AlphaFold2 (AF2) (*Jumper et al., 2021a*; *Tunyasuvunakool et al., 2021*; *Pereira et al., 2021*), which achieved remarkably accurate *de novo* structure prediction. Upon examining the list of CASP14 targets and corresponding models, we found that AF2 modeled the multidrug transporter LmrP (target T1024) in multiple conformations, two of which were individually consistent with published experimental data (*Jumper et al., 2021b*; *Del Alamo et al., 2021a*; *Debruycker et al., 2020*; *Martens et al., 2016*; *Masureel et al., 2014*). This observation stimulated the question of whether such performance can be duplicated for other membrane proteins. At its essence, this question centers on whether AF2 can sample the conformational landscape in the minimum energy basin. Here, we investigate this hypothesis using a benchmark set of topologically diverse transporters and GPCRs. Our results demonstrate that reducing the depth of the input MSAs is often conducive to the generation of accurate models in multiple conformations by AF2, suggesting that the algorithm's outstanding predictive performance can be extended to sample alternative structures of the same target. For most proteins considered, we report a striking correlation between the breadth of structures predicted by AF2 and the corresponding cryo-EM and/or X-ray crystal structures. Finally, we propose a modeling pipeline for researchers interested in sampling alternative conformations of specific membrane proteins, which we apply to the structurally unknown GPR114/ADGRG5 adhesion GPCR as an example.

## Results and discussion

The default three-stage AF2 pipeline consists of (1) querying of sequence databases and generation of an MSA, (2) inference of structure via a neural network using a randomly resampled subset of this MSA containing up to 5120 sequences, which is repeated three times (a process termed 'recycling'), and (3) resolution of steric clashes and bond geometry using a constrained all-atom molecular dynamics simulation. The neural networks used for prediction were trained on all structures deposited in the Protein Data Bank (PDB) on or before April 30, 2018 (*Jumper et al., 2021a*). Therefore, by necessity, this study is restricted to proteins whose structures were absent from the PDB before this date and have since been determined at atomic resolution in two or more conformations. We selected five transporters that not only met these criteria but also reflected a range of transport mechanisms characterized in the literature (*Drew and Boudker, 2016*), including rocking-bundle (LAT1, *Yan et al., 2019*; *Yan et al., 2021*; ZnT8, *Xue et al., 2020*), rocker-switch (MCT1, *Wang et al., 2021*; STP10, *Bavnhøj et al., 2021*), and elevator (ASCT2, *Garibsingh et al., 2021*; *Garaeva et al., 2019*). We also included three GPCRs, which were distributed across classes A (CGRPR, *Liang et al., 2020*; *Josephs et al., 2021*), B1 (PTH1R, *Ehrenmann et al., 2018*; *Zhao et al., 2019*), and F (FZD7, *Xu et al., 2021*; to serve as points of comparison, we used the active conformation of FZD7 and the inactive conformation of the nearly identical FZD4, *Yang et al., 2018*).

### AF2 generates multiple conformations of all eight target proteins

The sequences of all targets were truncated at the N- and C-termini to remove large soluble and/or intrinsically disordered regions which represent a challenge for AF2 (see *Methods*). The structures were then predicted using the default AF2 structure prediction pipeline in the absence of templates. However, the resulting models were largely identical to one another and failed to shed light on the target protein's conformational space. To diversify the models generated by AF2, we reduced the number of recycles to one and restricted the depth of the randomly subsampled MSAs to contain as few as 16 sequences. To sample the conformational landscape more exhaustively, we generated 50 models of each protein for each MSA size, while skipping the final MD simulation to reduce the

pipeline's total computational cost. For the targets, each model's similarity to the experimental structures was quantified using template modeling (TM) score (*Zhang and Skolnick, 2004*; *Zhang and Skolnick, 2005*; *Xu and Zhang, 2010*), a metric ranging from 0 to 1, which indicates how well the two backbone ($C_\alpha$ atoms) structures superimpose over one another (higher values corresponding to greater similarity; *Figure 1A*).

Accurate models of all eight protein targets were obtained for at least one conformation (TM score ≥0.9), consistent with published performance statistics (*Figure 1B*). MSAs with hundreds or thousands of sequences were generally observed to engender tighter clustering in conformations specific to each protein. Decreasing the depth of the subsampled MSAs, by contrast, appeared to promote the generation of alternative conformations in most proteins. The increased diversity coincided with the generation of misfolded or outlier models. However, unlike the models of interest that resembled experimentally determined structures, misfolded models virtually never coclustered and could thus be identified and excluded from further analysis (example shown in *Figure 1—figure supplement 1*). Increasing the depth of subsampled MSAs had the desirable effect of eliminating these models, while also limiting the extent to which alternative conformations were sampled. Thus, our results revealed a delicate balance that must be achieved to generate models that are both diverse and natively folded. No general pattern was readily apparent regarding the ideal MSA depth required to achieve this balance, even when accounting for sequence length of the target (*Figure 1—figure supplement 2*).

One target, MCT1, was exclusively modeled by AF2 in either inward-facing (IF) or fully occluded conformations; over 99% of the models had TM scores of ≥0.9 and <0.9 to the IF and outward-facing (OF) structures, respectively, regardless of MSA depth. Therefore, we investigated the effect of providing templates of homologs in exclusively OF conformations alongside MSAs of various sizes (see *Methods* for details on template selection). Accurate OF models were obtained only with MSAs containing 16–32 sequences and constituted a minor population in an ensemble dominated by IF models. Thus, the generation of large numbers of models appeared to be necessary to yield intermediate conformations of interest. Similar results were observed when we modeled PTH1R using either inactive or active templates, as well as LAT1 using either OF or IF templates (*Figure 1—figure supplement 3*), further indicating that the information content provided by the templates diminishes as the depth of the MSA increases.

Overall, these results demonstrate that both conformations of all eight protein targets could be predicted with AF2 to high accuracy (TM score ≥0.9) by using MSAs that are far shallower than the default. However, because the optimal MSA depth and choice of templates varied for each protein, these parameters need to be explored for conformational sampling of a particular target.

## Predicted conformational fluctuations correlate with implied conformational dynamics

To further investigate the structural heterogeneity predicted by these models, we calculated each residue's $C_\alpha$ atom distance between the two superimposed experimental structures, as well as each residue's root mean square fluctuation (RMSF) among all 50 models following structure-based alignment (*Figure 2*). Correlation between these two measures was observed in most cases and was notable for ASCT2, LAT1, CGRPR, and MCT1 with templates ($R^2 \geq 0.75$). The exception was MCT1 without templates, which was likely due to a lack of conformational diversity among the sampled models. The inclusion of templates restored this correlation in MCT1 but contributed negligibly to those of PTH1R and LAT1 (*Figure 2—figure supplement 1*). The correlation demonstrates that predicted flexibility by AF2 is related to the protein's dynamics inferred from the experimental structures. In contrast with a recent preprint (*Saldaño et al., 2021*), the predicted flexibility values failed to correlate with their pLDDT values, which reflect the confidence of the AF2 prediction of each residue's local environment (*Mariani et al., 2013*).

## Distributions of predicted models relative to the experimental structures

Visual examination suggested that many of the predicted models fall 'in between' the two experimentally determined conformations (example shown in *Figure 3A*). Furthermore, certain structural features expected to be conformationally heterogeneous, such as long loops, appeared to be nearly identical across these models. Both observations raised questions about the relationship between

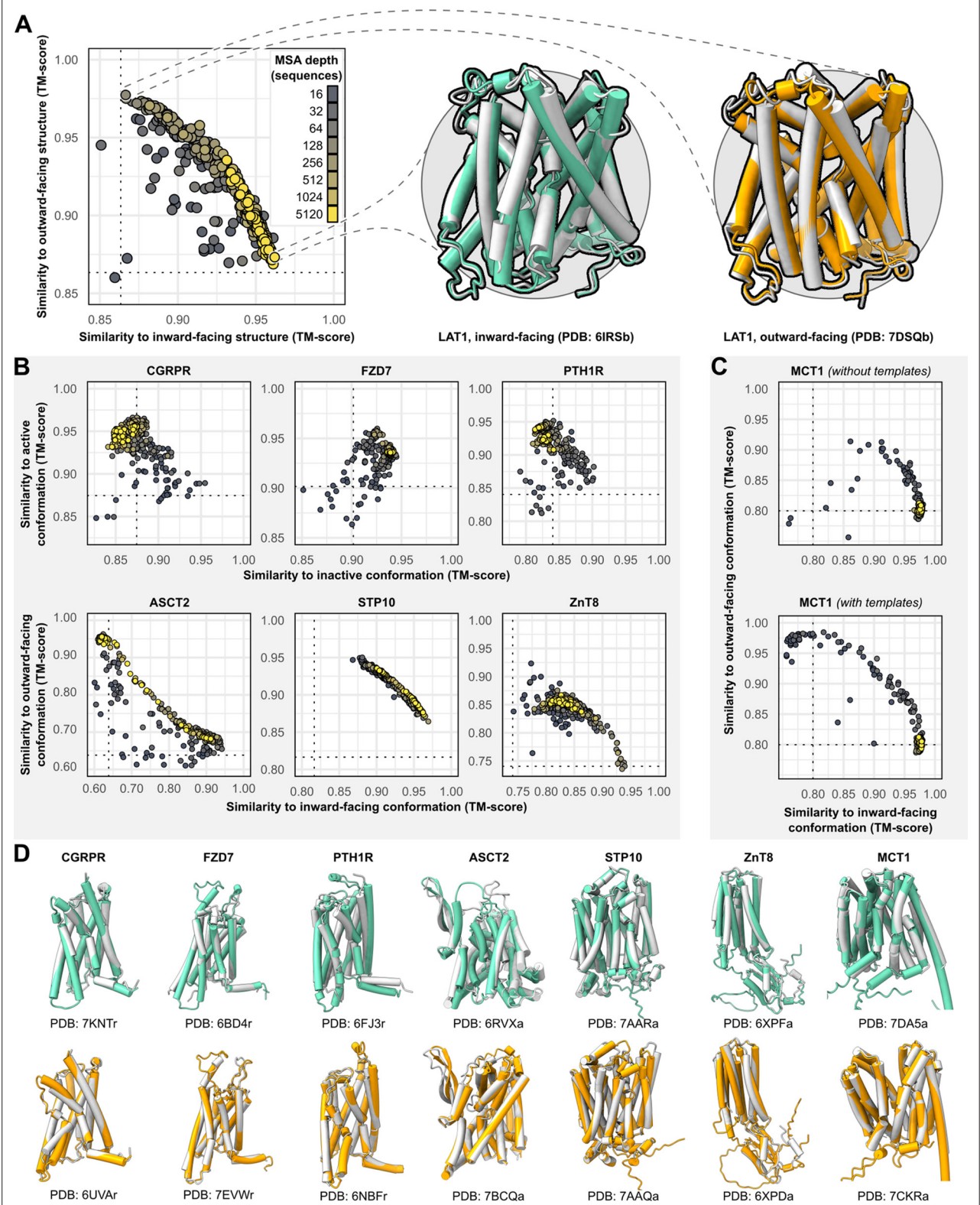

**Figure 1.** Alternative conformations of transporters and G-protein-coupled receptors (GPCRs) can be predicted by AlphaFold2 (AF2). (**A**) Representative models of the transporter LAT1 in inward-facing (IF) and outward-facing (OF) conformations. Experimental structures shown in gray and models shown in colors. (**B**) Comparison of AF2 models with inactive/active or IF/OF experimental structures as a function of multiple sequence alignment (MSA) depth for GPCRs (top) and transporters (bottom), respectively. All models shown here were generated without templates. Dashed lines indicate the template

*Figure 1 continued on next page*

*Figure 1 continued*

modeling (TM) score between experimental structures and are shown for reference. (**C**) Supplementing shallow MSAs with OF templates allows AF2 to predict the OF conformation of MCT1. (**D**) Experimental structures superimposed over models with the greatest TM scores. Inactive/IF and active/OF cartoons shown on the top and bottom in teal and orange, respectively.

The online version of this article includes the following figure supplement(s) for figure 1:

**Figure supplement 1.** Example principal component analysis (PCA) of ASCT2 models generated by AlphaFold2 (AF2) containing outlier models.

**Figure supplement 2.** Conformational homogeneity as a function of multiple sequence alignment (MSA) depth.

**Figure supplement 3.** Templates contribute to conformational sampling only when shallow multiple sequence alignments (MSAs) are provided.

**Figure supplement 4.** Protein targets with one conformation in the training set cannot be predicted in the alternative conformation.

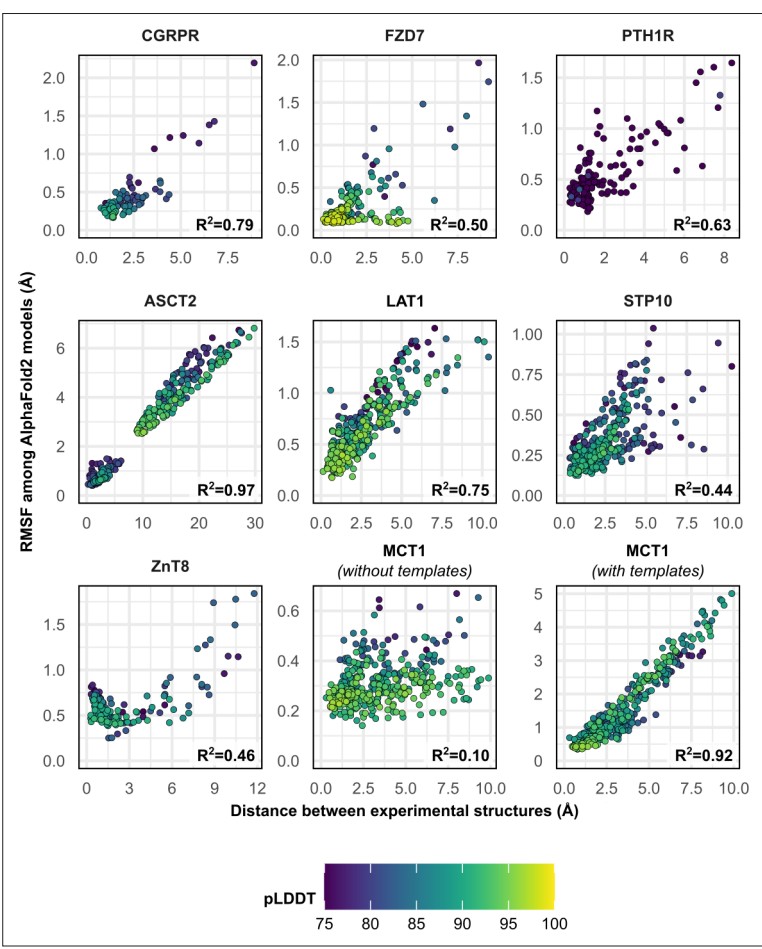

**Figure 2.** Comparison between the movement undergone by each $C_\alpha$ atom between the two superimposed experimental structures and their root mean square fluctuation (RMSF) values among AlphaFold2 (AF2) models. Residues with low confidence (pLDDT ≤75) were omitted from this plot for all proteins except PTH1R. Multiple sequence alignment (MSA) sizes of 128 sequences were used for all predictions, except for MCT1 with templates, which instead used 32 sequences to capture the outward-facing (OF) conformation. pLDDT refers to each residue's predicted accuracy, with a value of 100 indicating maximum confidence.

The online version of this article includes the following figure supplement(s) for figure 2:

**Figure supplement 1.** Comparison between the movement undergone by each $C_\alpha$ atom between the two superimposed experimental structures of PTH1R and LAT1 and their root mean square fluctuation (RMSF) values among AlphaFold2 (AF2) models generated with templates.

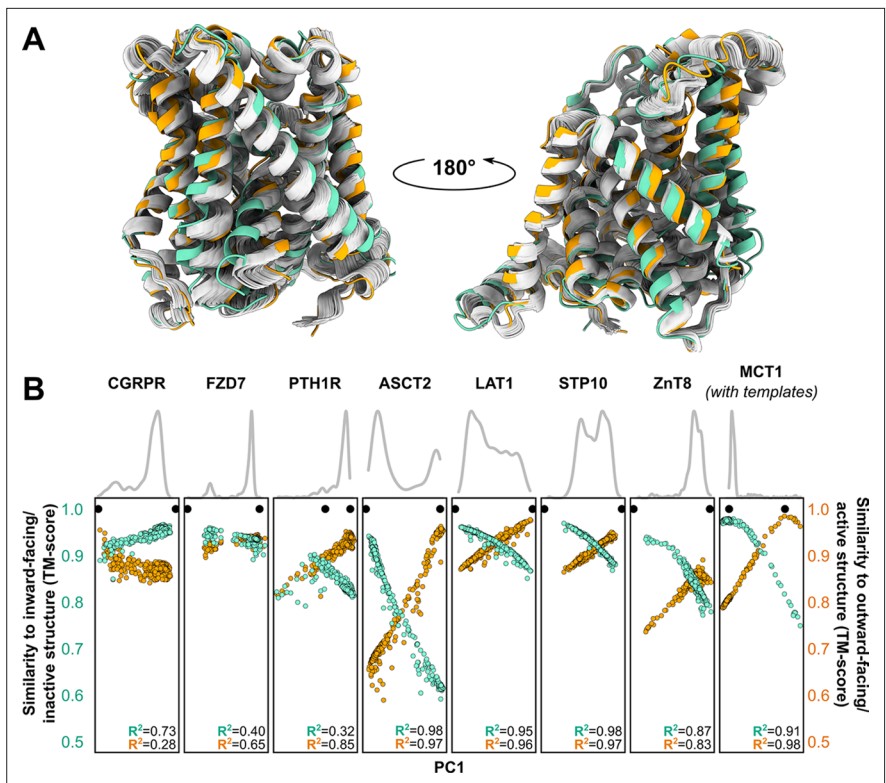

**Figure 3.** Distinct conformations can be delineated using principal component analysis (PCA). (**A**) Conformational heterogeneity in AlphaFold2 (AF2) models of LAT1. Experimental inward-facing (IF) and outward-facing (OF) conformations shown in teal and orange, respectively, while the gallery of AF2 models generated using 128 sequences are shown in gray. (**B**) Distribution of AF2 models generated using multiple sequence alignments (MSAs) with 32 or more sequences across the first principal component (PC1) following PCA (gray traces). Scatter plots comparing each model's position along PC1 and its structural similarity to experimentally determined structures. Teal: similarity to IF (transporters) or inactive (G-protein-coupled receptors, GPCRs) conformation. Orange: similarity to OF (transporters) or active (GPCRs) conformation. Each model is shown twice, once in teal and once in orange. Native structures are shown as black dots.

The online version of this article includes the following figure supplement(s) for figure 3:

**Figure supplement 1.** Models sampled by AlphaFold2 (AF2) in multiple conformations cannot be fully explained by linear interpolation of two end structures.

**Figure supplement 2.** Example predictions of the adhesion G-protein-coupled receptor (aGPCR) GPR114/ADGRG5.

the diversity of the predicted models and the breadth of the conformational ensembles bracketed by the experimental structures. To quantitatively place the predicted conformational variance in the context of the experimentally determined structures, we used principal component analysis (PCA), which reduces the multidimensional space to a smaller space representative of the main conformational motions. In our benchmark set, the first principal component (PC1) captured 64.9% ± 16.1% of the structural variations among the models generated using MSAs with 32 or more sequences (*Figure 3B*), while comparison of PC1/PC2 values suggested that the predicted dynamics deviate from simple interpolation of two end states (*Figure 3—figure supplement 1*). The experimental structures virtually always occupied well-defined extreme positions. In every case, a correlation was evident between each model's PC1 values and their TM scores. Indeed, the models with the most extreme PC1 values were also among the most accurate: average TM scores were 0.94 for the top 1, top 3, and top 10 PC1 models, and Pearson's correlation coefficients between PC1 and TM scores of the ensemble of models exceeded 0.8 for all transporters in this dataset. Moreover, the experimental structures virtually always flanked the AF2 models along PC1. The exception, PTH1R, was determined in a partially inactive and active conformation (*Zhao et al., 2019*), suggesting that models extending

beyond the former state along PC1 may represent the fully inactive conformation. Therefore, these results indicate that accurate representative models of conformations of interest can be selected from the extreme positions along PC1.

## Limited conformational sampling is observed for proteins with structures in the AF2 training set

A follow-up question centers on whether this strategy can yield similar results for proteins with one conformation present in the AF2 training set. We investigated this question using four membrane proteins with two experimentally determined conformations, at least one of which was included in the AF2 training set: the class A GPCR CCR5 (*Zheng et al., 2017*; *Zhang et al., 2021*), the serotonin transporter SERT (*Coleman et al., 2016*; *Coleman et al., 2019*), the multidrug transporter PfMATE (*Tanaka et al., 2013*; *Zakrzewska et al., 2019*), and the lipid flippase MurJ (*Kuk et al., 2017*; *Kuk et al., 2019*). Using the template-free prediction pipeline outlined above, we determined the resultant models' similarity to the structures included in and absent from the training set. Unlike the results presented above, the majority of the transporter models generated this way were more similar to the conformation present in the training set than the conformation absent from the training set (i.e., their TM scores were greater; *Figure 1—figure supplement 4*). The conformational diversity of these models, including those generated using shallow MSAs, was far more limited than what was generally observed for the proteins discussed above, with the exception of MCT1 (*Figure 1—figure supplement 2*). Although conformational diversity was demonstrated to a limited extent by the generation of occluded models of MurJ and PfMATE, none of the models observed adopted the alternative conformer. By contrast, while models of CCR5 were less biased toward the training set conformation, deep MSAs reduced conformational diversity. This divergence in performance may stem from the composition of the AF2 training set, which featured the structures of many active GPCRs but no structures, for example, of IF MATEs (*Claxton et al., 2021*).

## Concluding remarks: proposed workflow and future directions

Our results indicate that the state-of-the-art *de novo* structural modeling algorithm AF2 can be manipulated to accurately model alternative conformations of transporters and GPCRs whose structures were not available in the training set. The use of shallow MSAs was instrumental to obtaining structurally diverse models in most proteins, and in one case (MCT1) accurate modeling of alternative conformations also required the manual curation of template structures. Thus, while the results presented here provide a blueprint for obtaining AF2 models of alternative conformations, they also argue against an optimal one-size-fits-all approach for sampling the conformational space of every protein with high accuracy. Indeed, whereas the DeepMind team reportedly required templates to obtain models of LmrP in an OF conformation (*Jumper et al., 2021b*), we found that this procedure was usually unnecessary. Accurate representatives of distinct conformers were generally obtainable with exhaustive sampling and could be identified by performing PCA and selecting models at the extreme positions of PC1. Nevertheless, prediction pipelines will likely require a combination of iterative fine-tuning specific to each target of interest followed by experimental verification to identify proposed conformers. Moreover, this approach showed limited success when applied to transporters whose structures were used to train AF2, hinting at the possibility that traditional methods may still be required to capture alternative conformers (*Crawley et al., 2011*; *Ollikainen et al., 2013*).

As a final verification of this proposed pipeline, we tested it on GPR114/ADGRG5, a class B2 adhesion GPCR whose structure has not been experimentally determined. The structural model deposited in the AF2 database, which likely depicts an active conformation that diverges from the structure of the homolog GPR97 (*Ping et al., 2021*), could be recapitulated by using deep MSAs. The use of shallow MSAs (≤64 sequences), by contrast, yielded a range of intermediate conformations distributed across three well-separated clusters (*Figure 3—figure supplement 2*). One of these clusters contains models with an orientation of TM6 and TM7 that fully occludes the orthosteric site and partially blocks the cytosolic pocket where G proteins bind. The physiological relevance of these proposed structural movements nonetheless requires experimental validation.

While these results reinforce the notion that AF2 can provide models to guide biophysical studies of conformationally heterogeneous membrane proteins, they represent a methodological 'hack', rather than an explicit objective built into the algorithm's architecture. Several preprints have provided

evidence that AF2, despite its accuracy, likely does not learn the energy landscapes underpinning protein folding and function (*Saldaño et al., 2021*; *Pak et al., 2021*; *Akdel et al., 2021*). Moreover, AF2 does not directly account for the lipid environment, which has been experimentally shown to bias the conformational equilibria of membrane proteins (*Martens et al., 2018*; *Muller et al., 2019*; *Immadisetty et al., 2019*). As our results show, the exploration of the conformational space is in part a byproduct of low sequence information provided for inference. Ultimately, they highlight the need for further development of artificial intelligence methods capable of learning the conformational flexibility intrinsic to protein structures.

## Materials and methods

### Overview of the prediction pipeline

Prediction runs were executed using AlphaFold v2.0.1 and a modified version of ColabFold (*Mirdita et al., 2021*) that is available at https://github.com/delalamo/af2_conformations, (*Del Alamo, 2021b* copy archived at swh:1:rev:d60db86886186e80622deaa91045caccaf4103d3). The pipeline used in this study differs from the default AF2 pipeline in several aspects. First, all MSAs were obtained using the MMSeqs2 server (*Steinegger and Söding, 2017*), rather than the default databases. Second, template search was disabled, except when explicitly performed with specific templates of interest (see below). Third, the number of recycles was set to one, rather than three by default. Finally, models were not refined following their prediction. This study utilized all 5 neural networks when predicting structures without templates, with 10 predictions per neural network per MSA size. The following residues were omitted from modeling: 1–131 and 401–461 of CGRPR, 1–247 of FZD7, 1–175 and 492–593 of PTH1R, and 1–49 of LAT1.

### MSA subsampling

MSA subsampling was carried out randomly by AF2, and depth values were controlled by modifying '*max_msa_clusters*' and '*max_extra_msa*' parameters prior to execution. The former parameter determines the number of randomly chosen sequence clusters provided to the AF2 neural network. The latter parameter determines the number of extra sequences used to compute additional summary statistics. Throughout this manuscript, we refer to the latter when describing the depth of the MSA used for prediction and set the former to half this value in all cases except when 5120 sequences were used, in which case we set the former to 512. No manual intervention was carried out to fine-tune the composition of these alignments.

### Template-based predictions

Templates were fetched using the MMSeqs2 server used by ColabFold. All templates were manually inspected, and those with structures similar to the desired conformation of interest were retained. The following templates were used: OF MCT1 with FucB (PDB 3O7Pa and 3O7Qa, 18% sequence identity calculated using Needleman Wunsch *Needleman and Wunsch, 1970*; *Madeira et al., 2019*); OF LAT1 with AdiC (3OB6a and 5J4Ib, 22%); IF LAT1 with b(0,+)AT1 (6LI9, 45%), KCC3 (6Y5Ra, 12%), GkApcT (6F34a, 24%), NKCC1 (6NPLa, 6PZTa, and 6NPHa, 12%), BasC (6F2Ga and 6F2Wa, 29%), and GadC (4DJIa and 4DJKb, 21%); active PTH1R with GCGR (6LMLr and 6VCBr, 30%) PAC1 (6M1I, 34%), and CRF1 (6P9X, 34%); inactive PTH1R with GLP1 (6LN2, 30%) and GCGR (5YQZ, 5XF1, and 5EE7, 28%). Template processing then proceeded as previously described, except that the parameter '*subsample_templates*' was set to True and the template similarity cutoff was reduced from 10% to 1%. Additionally, as only 2 of the 5 AF2 neural networks were parametrized to use templates, each of these 2 neural networks generated 25 models in order to arrive at 50 total models per MSA depth.

### Structural analysis

TM scores were calculated using TM align (*Zhang and Skolnick, 2005*). PCA and RMSF calculations were carried out in CPPTRAJ (*Roe and Cheatham, 2013*). Loop residues were omitted from PCA.

## Acknowledgements

This study was funded by the National Institutes of Health (HSM: GM 128087) and the Deutsche Forschungsgemeinschaft (DFG, German Research Foundation) through CRC 1423, project number

421152132, subproject Z04. The authors would like to thank Dr. John Jumper for explaining how the DeepMind team predicted the structure of LmrP in CASP14 and Mr. Taylor Jones for helpful discussions on modeling MCT1 without templates.

## Additional information

### Funding

| Funder | Grant reference number | Author |
|---|---|---|
| National Institutes of Health | GM 128087 | Hassane S Mchaourab |
| Deutsche Forschungsgemeinschaft | CRC 1423, project number 421152132, subproject Z04 | Jens Meiler |

The funders had no role in study design, data collection, and interpretation, or the decision to submit the work for publication.

### Author contributions

Diego del Alamo, Conceptualization, Data curation, Formal analysis, Methodology, Software, Validation, Visualization, Writing – original draft, Writing – review and editing; Davide Sala, Conceptualization, Data curation, Formal analysis, Investigation, Methodology, Software, Validation, Visualization, Writing – original draft, Writing – review and editing; Hassane S Mchaourab, Jens Meiler, Conceptualization, Funding acquisition, Project administration, Resources, Supervision, Writing – review and editing

### Author ORCIDs

Diego del Alamo http://orcid.org/0000-0003-1757-9971
Davide Sala http://orcid.org/0000-0002-3900-0011
Hassane S Mchaourab http://orcid.org/0000-0002-5673-0980
Jens Meiler http://orcid.org/0000-0001-8945-193X

### Decision letter and Author response

Decision letter https://doi.org/10.7554/eLife.75751.sa1
Author response https://doi.org/10.7554/eLife.75751.sa2

## Additional files

### Supplementary files

• Transparent reporting form

### Data availability

All scripts and data presented in this study are made available for download at https://github.com/delalamo/af2_conformations, (copy archived at swh:1:rev:d60db86886186e80622deaa91045caccaf4103d3).

The following dataset was generated:

| Author(s) | Year | Dataset title | Dataset URL | Database and Identifier |
|---|---|---|---|---|
| Del Alamo D | 2022 | Sampling alternative conformational states of transporters and receptors with AlphaFold2 | https://github.com/delalamo/af2_conformations | GitHub, conformations |

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
