## [Editor Report]

del Alamo and colleagues illustrate that restricting the depth of the input multiple sequence alignment allows AlphaFold2 to predict diverse conformational ensembles of transporters and receptors, as opposed to single static models reflecting individual states. Although they are limited to a small number of test cases of membrane proteins, the examples are of interest to members of the community. This work presents a validation of a simple approach that may be applicable to all proteins and is thus an exciting advance that is expected to be of broad interest.

---

## [Decision Letter]

**Decision letter after peer review:**

Thank you for submitting your article "Sampling the conformational landscapes of transporters and receptors with AlphaFold2" for consideration by *eLife*. Your article has been reviewed by 3 peer reviewers, including Janice L Robertson as the Reviewing Editor and Reviewer #1, and the evaluation has been overseen by a Reviewing Editor and Kenton Swartz as the Senior Editor.

In general, the reviewers were enthusiastic about your finding that reducing the size of the sequence alignments input into AlphaFold2 increases conformational diversity for predictions of transporter membrane protein folds. While the test set is small, the validation provided is convincing, and the results are likely to be broadly useful to many who study protein conformational changes. However, it was found that many details were lacking about the methods, which will limit the ability for others to reproduce these findings or advance this approach further. With that, the following revisions are required in order to describe the methods in appropriate detail and increase the quantitative presentation of the analysis. In addition, it is important to temper general claims made throughout the paper to acknowledge these findings are based on a small test set of proteins. Since these essential revisions focus mainly on writing with some minor additions to the analysis, we expect that these changes will be tractable within a reasonable time frame.

Essential revisions:

1) Elaboration of the methods used. Additional details are needed in order to be able to evaluate the validity and reproducibility of the approach. Specifically,

– Include a brief description of the AF2 protocol and each point at which variability is introduced, i.e. by sequence alignment, template choice or recycling.

– The alignments used to develop the models should be provided. Specific details on how the visual inspection of the alignments guided their refinement should also be included. What is padding of the MSAs? How were the MSAs trimmed from 5120 to 32? What was the distribution of lengths of the sequences included? What were the sequences that ended up being included and what was the sequence diversity in the sets used?

– Earlier in the paper, it is stated that loops were removed from the sequence alignments. However, they are later discussed as being generated in the models. Provide more details about when the loops were included in the structure prediction.

– For some of the targets, the template-based modeling clearly improved sampling of various conformations and for others it didn't. How were the template selected for the template-based modeling?

– What are the PDBs used in the structural analysis? These should be listed explicitly in the pertinent figure and in the methods.

– Define pLDDT.

– What does "eliminating postprocessing with OpenMM" constitute?

– How were misfolded models identified? Providing a reference is not sufficient here.

– To address the predictive power of this approach please clarify which models were used for the PCA. Were the principal components computed from low-MSA AlphaFold2 predictions only, rather than from the large-MSA AF2 predictions, which would make the point moot since the PC reflect the range of conformational changes observed in multiple models, not a subset. Previous studies (Bahar and colleagues) suggested that PCA allows for prediction, but that PC1 is not always the useful component and so the question arises of how to select the correct PC to make the prediction?

2) Additional analysis:

– Analysis of the model accuracy with alignment quality. How do the current results depend on alignment quality and diversity? Which sequences are included in the 32, and how do your findings depend on this selection?

– Analysis of the model accuracy with sequence length. While sequence information is examined, the authors say that no general pattern was apparent regarding the ideal MSA depth. Yet, a more common strategy, namely, to compare sequence sets using a factor related to the length (L) of the protein (or perhaps the core of the protein being modeled) may reveal more. Indeed, by reducing the dataset to 32 sequences, only the longest proteins were starting to include misfolded examples. Overall, it would be more straightforward to compare models built with, e.g. L*2, L/2 and L/5 sequences. While this requires building additional models it would also provide a clearer outcome and strategy that future users could follow. A bonus may be that it would reduce the chances of misfolded models that need to be filtered out. At the minimum, the authors should reframe the data they have as a function of each protein's length.

– Analysis of template usage. What were the templates used? Was the performance of AF2 dependent on the sequence similarity between the template(s) and the target?

– Quantification of conformations. There are many occasions where the discussion of structural similarities/differences are qualitative, e.g. “virtually every transporter model superimposed nearly perfectly with the training set conformation, and none resembled the alternative conformation”. This statement should be accompanied by quantitative data. Furthermore, the different known conformational states, i.e. IF, OF and occluded, require a quantitative definition to support statements like “One target, MCT1, was exclusively modeled by AF2 in either IF or fully occluded conformations regardless of MSA depth. Notably, these results closely parallel those reported by DeepMind during their attempt to model multiple conformations of LmrP in CASP14.”.

– Along these lines, it is reported that conformational variability is not obtained by the targets that were included in the AF2 training set, and yet the extent of conformational diversity appears similar to that analysis presented in Figure 1. For example, MurJ appears to show the same degree of conformational sampling with 32 sequences as for ASCT2. A more objective analysis of the conformational sampling is required to define the dynamic range explored by the structural conformations, especially since some of the endpoint structures are quite similar to each other.

3) Please respond to and address the additional recommendations provided by the reviewers.

*Reviewer #1 (Recommendations for the authors):*

1. The conformational ensemble from AF2 appears to move along certain structural paths in the different analyses. How does this compare to a linear interpolation between the endpoint structures?

2. In Figure 1, it is recommended that the axes of Figure 1B be scaled similarly to the format used in Figure S3 since the experimental TM-score differences are quite different between the different proteins. Aside from ASCT2, and potentially MCT1 with templates, the dynamic range of the conformational change appears to be minimal, but this may just be difficult to see due to the current plot format. In addition, move Figure S1 into this main figure to allow the reader to discern the structural and conformational variability in this test set. Finally, please add all of the pdbs used for the experimental comparison structures, both in this figure, and in the methods.

3. Is the term “ground truth structures” referring to the crystal structures or other experimental structures? Please change this term as experimental structures do not correspond to a “truth” but is a physically accessible conformational state of the protein under those experimental conditions.

*Reviewer #2 (Recommendations for the authors):*

1. First, I would disagree with the title of Figure S5 and the beginning of the corresponding title, which seem to be categorical about the lack of exploration of alternate conformations for these examples, but then somewhat contradict the rest of the paragraph, where it is explained that some cases (MurJ and CCR5) behave differently from the others. These discrepancies should be resolved.

2. Second, I think it would be of value for the readership to mention that no function is included to describe the membrane in these modelling processes – even when the lipids themselves may be critical to shift these conformational equilibria. This observation actually makes the authors’ findings all the more remarkable, but also perhaps harder to interpret.

*Reviewer #3 (Recommendations for the authors):*

1. Change Lat1 -> LAT1

2. The following statement is unclear and should be elaborated. “Several preprints have provided evidence that AF2, despite its accuracy, likely does not learn the energy landscapes underpinning protein folding and function39,53,54. We believe that our results bolster these conclusions and highlight the need for further development of artificial intelligence methods capable of learning the conformational flexibility intrinsic to protein structures.”

[Editors’ note: further revisions were suggested prior to acceptance, as described below.]

Thank you for resubmitting your work entitled “Sampling the conformational landscapes of transporters and receptors with AlphaFold2” for further consideration by *eLife*. Your revised article has been evaluated by Kenton Swartz (Senior Editor) and a Reviewing Editor.

The manuscript has been improved but there are some remaining issues that need to be addressed, as outlined below:

1. The reviewers found that the current title may lead the reader to misinterpretation. An alternate title “Sampling alternative conformational states of transporters and receptors with AlphaFold2” is more appropriate and should be adopted.

2. The findings in Figure 1 Suppl2, that the number of sequences isn’t correlated with an increase in conformational homogeneity, and displays erratic dependence for some proteins (especially ASCT2, Lat1 and STP10), are surprising. Consequently, it seems necessary to alter some statements in the manuscript, accordingly. For example, in the abstract: “reducing the depth of the input MSAs is conducive to the generation of accurate models in multiple conformations by AF2”.

---

## [Author Response]

Essential revisions:1) Elaboration of the methods used. Additional details are needed in order to be able to evaluate the validity and reproducibility of the approach. Specifically,– Include a brief description of the AF2 protocol and each point at which variability is introduced, i.e. by sequence alignment, template choice or recycling.

(0.1.1) We have added several sentences introducing the general pipeline at the beginning of “Results and Discussion” (references omitted for clarity):

“The default three-stage AF2 pipeline consists of (1) querying of sequence databases and generation of an MSA, (2) inference via a neural network using a randomly resampled subset of this MSA containing up to 5120 sequences, which is repeated a total of three times (a process termed “recycling”), and (3) resolution of steric clashes and bond geometry using a constrained all-atom molecular dynamics simulation. The neural networks used for prediction were trained on all structures deposited in the protein data bank (PDB) on or before April 30th, 2018^11^. Therefore, by necessity, this study is restricted to proteins whose structures were absent from the PDB before this date and have since been determined at atomic resolution in two or more conformations.”

Additionally, to reflect both this comment and other comments made by Reviewers (see below), the following paragraph was modified to detail how this pipeline was changed:

“However, the resulting models were largely identical to one another and failed to shed light on the target protein’s conformational space. To diversify the models generated by AF2, we reduced the number of recycles to one and restricted the depth of the randomly subsampled MSAs to contain as few as 16 sequences. To sample the conformational landscape more exhaustively, we generated fifty models of each protein for each MSA size, while skipping the final MD simulation to reduce the pipeline's total computational cost.”

Finally, we have rewritten the Methods section to accommodate additional detail for the prediction pipeline described herein:

“Overview of the prediction pipeline

Prediction runs were executed using AlphaFold v2.0.1 and a modified version of ColabFold^54^ that is available at www.github.com/delalamo/af2_conformations. The pipeline used in this study differs from the default AF2 pipeline in several respects. First, all MSAs were obtained using the MMSeqs2 server^55^, rather than the default databases. Second, template search was disabled, except when explicitly performed with specific templates of interest (see below). Third, the number of recycles was set to one, rather than three by default. Finally, models were not refined following their prediction. Unlike the ColabFold pipeline, however, this study utilized all five neural networks when predicting structures without templates, with ten predictions per neural network per MSA size. Additionally, as only two of the five neural networks can use templates to supplement MSAs for structure prediction, they each performed twenty-five predictions per MSA size when performing template-based predictions. The following residues were omitted from modeling: 1-131 and 401-461 of CGRPR, 1-247 of FZD7, 1-175 and 492-593 of PTH1R, and 1-49 of LAT1.”

“MSA subsampling

MSA subsampling was carried out randomly by AF2, and depth values were controlled by modifying "max_msa_clusters" and "max_extra_msa" parameters prior to execution. The former parameter determines the number of randomly chosen sequence clusters provided to the AF2 neural network. The latter parameter determines the number of extra sequences used to compute additional summary statistics. Throughout this manuscript, we refer to the latter when describing the depth of the MSA used for prediction and set the former to half this value in all cases except when 5120 sequences were used, in which case we set the former to 512. No manual intervention was carried out to fine-tune the composition of these alignments.”

“Template-based predictions

Templates were fetched using the MMSeqs2 server used by ColabFold. All templates were manually inspected, and those with structures similar to the desired conformation of interest were retained. The following templates were used: outward-facing MCT1 with FucB (PDB 3O7Pa and 3O7Qa, 18% sequence identity calculated using Needleman Wunsch^56,57^); outward-facing LAT1 with AdiC (3OB6a and 5J4Ib, 22%); inward-facing LAT1 with b(0,+)AT1 (6LI9, 45%), KCC3 (6Y5Ra, 12%), GkApcT (6F34a, 24%), NKCC1 (6NPLa, 6PZTa, and 6NPHa, 12%), BasC (6F2Ga and 6F2Wa, 29%), and GadC (4DJIa and 4DJKb, 21%); active PTH1R with GCGR (6LMLr and 6VCBr, 30%) PAC1 (6M1I, 34%), and CRF1 (6P9X, 34%); inactive PTH1R with GLP1 (6LN2, 30%) and GCGR (5YQZ, 5XF1, and 5EE7, 28%). Template processing then proceeded as previously described, except that the parameter “subsample_templates” was set to True and the template similarity cutoff was reduced from 10% to 1%. Additionally, as only two of the five AF2 neural networks were parametrized to use templates, each of these two neural networks generated twenty-five models in order to arrive at fifty total models per MSA depth.”

“Structural analysis

TM-scores were calculated using TM-align^33^. Principal component analysis and RMSF calculations were carried out in CPPTRAJ^58^. Loop residues were omitted from PCA.”

– The alignments used to develop the models should be provided. Specific details on how the visual inspection of the alignments guided their refinement should also be included. What is padding of the MSAs? How were the MSAs trimmed from 5120 to 32? What was the distribution of lengths of the sequences included? What were the sequences that ended up being included and what was the sequence diversity in the sets used?

(0.1.2) We have clarified in “Results and Discussion” that all manipulation of the MSA was performed automatically by the AF2 program. Our pipeline in this study simply reduced the depth of the subsampled MSAs without manual tuning the content. Our changes are described above in response to the previous comment. Additionally, we have edited the text in “Abstract” to emphasize this point:

“Whereas models generated using the default AF2 pipeline are conformationally homogeneous and nearly identical to one another, reducing the depth of the input multiple sequence alignments (MSAs) by stochastic subsampling led to the generation of accurate models in multiple conformations.”

We note that this critical point is reinforced in the “Methods” section (see 0.1.1 above). Finally, throughout the text, we have modified the text to emphasize that AF2 randomly determined the composition of the subsampled MSAs used for structure prediction (for example, by removing the word “padding” on page 3). We therefore believe that details regarding the optimal composition of the MSAs required to obtain alternative conformations is beyond the scope of this publication. Moreover, to our knowledge, AF2 currently does not support a way to extract the subsampled MSA used for the prediction.

– Earlier in the paper, it is stated that loops were removed from the sequence alignments. However, they are later discussed as being generated in the models. Provide more details about when the loops were included in the structure prediction.

(0.1.3) We have clarified the text to mention that sequence truncation was limited to the N- and C-termini:

“The sequences of all targets were truncated at the N- and C-termini to remove large soluble domains attached to the membrane proteins and/or intrinsically disordered regions.”

We have also modified “Methods” to account for the removal of loop residues during PCA.

– For some of the targets, the template-based modeling clearly improved sampling of various conformations and for others it didn't. How were the template selected for the template-based modeling?

(0.1.4) We have added the relevant text to the manuscript:

“Therefore, we investigated the effect of providing templates of homologs in exclusively OF conformations alongside MSAs of various sizes (see Methods for details on template selection). Accurate OF models were obtained only with MSAs containing 16 to 32 sequences and constituted a minor population in an ensemble dominated by IF models.”

Additional changes were introduced in Methods (see 0.1.1 above).

– What are the PDBs used in the structural analysis? These should be listed explicitly in the pertinent figure and in the methods.

(0.1.5) We have modified Figures 1 and the caption of Figure 1 —figure supplement 4 (formerly Figure S5) to include all relevant PDB accession codes. Some of the modifications made to Figure 1 are described below.

– Define pLDDT.

(0.1.6) We have modified the relevant section in “Results and Discussion” to include a definition:

“In contrast with a recent preprint^35^, the predicted flexibility values failed to correlate with their pLDDT values, which reflect the confidence of the AF2 prediction of each residue’s local environment^36^.”

– What does "eliminating postprocessing with OpenMM" constitute?

(0.1.7) We have modified the text to provide clarification (see 0.1.1 above).

– How were misfolded models identified? Providing a reference is not sufficient here.

(0.1.8) In light of both comments made by the Reviewers below, as well as new calculations carried out in response to these comments, we have revised this paragraph to include both new quantitative descriptions of results and a PCA-based approach for identifying and removing misfolded outlier models:

“Accurate models of all eight protein targets were obtained for at least one conformation (TM-score ≥0.9), consistent with published performance statistics (Figure 1B). MSAs with hundreds or thousands of sequences were generally observed to engender tighter clustering in conformations specific to each protein. Decreasing the depth of the subsampled MSAs, by contrast, appeared to promote the generation of alternative conformations. The increased diversity coincided with the generation of misfolded or outlier models. However, unlike the models of interest that resembled experimentally determined structures, misfolded models virtually never co-clustered and could thus be identified and excluded from further analysis (example shown in Figure 1 —figure supplement 1). Increasing the depth of subsampled MSAs had the desirable effect of eliminating these models, while also limiting the extent to which alternative conformations were sampled. Thus, our results revealed a delicate balance that must be achieved to generate models that are both diverse and natively folded. No general pattern was readily apparent regarding the ideal MSA depth required to achieve this balance, even when accounting for sequence length of the target (Figure 1 —figure supplement 2).”

We have also replaced the previous supplemental Figure (formerly name Figure S2, now Figure 1 —figure supplement 1), which only shows an isolated example of an outlier model, with an example principal component analysis of ASCT2 that we encountered during this study. This was the same approach we used when generating Figure 3 and Figure 3 —figure supplement 2 (formerly called Figure S6).

– To address the predictive power of this approach please clarify which models were used for the PCA. Were the principal components computed from low-MSA AlphaFold2 predictions only, rather than from the large-MSA AF2 predictions, which would make the point moot since the PC reflect the range of conformational changes observed in multiple models, not a subset. Previous studies (Bahar and colleagues) suggested that PCA allows for prediction, but that PC1 is not always the useful component and so the question arises of how to select the correct PC to make the prediction?

(0.1.9) We have clarified in the text that dimensionality reduction using PCA was carried out on all models obtained for all MSA depth values:

“In our benchmark set, the first principal component (PC1) captured 64.9±16.1% of the structural variations among all the models generated using MSAs with 32 or more sequences (Figure 3B).”

We also edited the caption of Figure 3:

We also added Figure 3 —figure supplement 1 showing both PC1 and PC2 to illustrate the limited interpretive power of PC1.

2) Additional analysis:– Analysis of the model accuracy with alignment quality. How do the current results depend on alignment quality and diversity? Which sequences are included in the 32, and how do your findings depend on this selection?

(0.2.1) We clarified in our text that the composition of the sequences being subsampled is decided entirely randomly by AF2 (see 0.1.2 above).

– Analysis of the model accuracy with sequence length. While sequence information is examined, the authors say that no general pattern was apparent regarding the ideal MSA depth. Yet, a more common strategy, namely, to compare sequence sets using a factor related to the length (L) of the protein (or perhaps the core of the protein being modeled) may reveal more. Indeed, by reducing the dataset to 32 sequences, only the longest proteins were starting to include misfolded examples. Overall, it would be more straightforward to compare models built with, e.g. L*2, L/2 and L/5 sequences. While this requires building additional models it would also provide a clearer outcome and strategy that future users could follow. A bonus may be that it would reduce the chances of misfolded models that need to be filtered out. At the minimum, the authors should reframe the data they have as a function of each protein's length.

(0.2.2) We appreciate this suggestion and have added Figure 1 —figure supplement 2 to compare structural variation as a function of MSA depth normalized by sequence length. We reran the protocol with a more comprehensive range of MSA depths in an attempt to identify a pattern. However, due to the small size of our test set, our results are inconclusive. Nevertheless, they have allowed us to make a clarifying point regarding the modeling of proteins in the training set (discussed below).

– Analysis of template usage. What were the templates used? Was the performance of AF2 dependent on the sequence similarity between the template(s) and the target?

(0.2.3) We have modified the “Methods” section to describe our protocol and have added Table S3 listing the templates used (see 0.1.4 above).

– Quantification of conformations. There are many occasions where the discussion of structural similarities/differences are qualitative, e.g. “virtually every transporter model superimposed nearly perfectly with the training set conformation, and none resembled the alternative conformation”. This statement should be accompanied by quantitative data. Furthermore, the different known conformational states, i.e. IF, OF and occluded, require a quantitative definition to support statements like “One target, MCT1, was exclusively modeled by AF2 in either IF or fully occluded conformations regardless of MSA depth. Notably, these results closely parallel those reported by DeepMind during their attempt to model multiple conformations of LmrP in CASP14.”.

(0.2.4) We have made several changes to the text as recommended:

“Accurate models of all eight protein targets were obtained for at least one conformation (TM-score > 0.9), consistent with published performance statistics.”

“One target, MCT1, was exclusively modeled by AF2 in either IF or fully occluded conformations; over 99% of the models had TM-scores of ≥0.9 and <0.9 to the IF and OF structures, respectively, regardless of MSA depth.”

“Overall, these results demonstrate that both conformations of all eight protein targets could be predicted with AF2 to high accuracy (TM-score ≥0.9) by using MSAs that are far shallower than the default.”

Additionally, in response to comments made by another Reviewer, we have removed the reference to the CASP14 result in this subsection of Results and Discussion.

– Along these lines, it is reported that conformational variability is not obtained by the targets that were included in the AF2 training set, and yet the extent of conformational diversity appears similar to that analysis presented in Figure 1. For example, MurJ appears to show the same degree of conformational sampling with 32 sequences as for ASCT2. A more objective analysis of the conformational sampling is required to define the dynamic range explored by the structural conformations, especially since some of the endpoint structures are quite similar to each other.

(0.2.5) We have modified the language accordingly:

“Unlike the results presented above, the majority of the transporter models generated this way were more similar to the conformation present in the training set than the conformation absent from the training set (i.e., their TM-scores were greater; Figure 1 —figure supplement 4).”

Additionally, we have plotted these results in Figure 1 —figure supplement 2 above to demonstrate their structural homogeneity relative to proteins absent from the training set (see 0.2.2 above).

Reviewer #1 (Recommendations for the authors):1. The conformational ensemble from AF2 appears to move along certain structural paths in the different analyses. How does this compare to a linear interpolation between the endpoint structures?

(1.1) This question dovetails with the recommendation outlined by the Editor above. We have therefore added a supplemental figure to show that the first principal component describes only a subset of the variation observed in the models (see 0.1.9 above).

This Figure is referenced in the following sentence in “Results and Discussion” subsection “Distributions of predicted models relative to the experimental structures”:

“In our benchmark set, the first principal component (PC1) captured 64.9±16.1% of the structural variations among the models generated using MSAs with 32 or more sequences (Figure 3B), while comparison of PC1/PC2 values suggested that the predicted dynamics deviate from simple interpolation of two end states (Figure 3 —figure supplement 1).”

2. In Figure 1, it is recommended that the axes of Figure 1B be scaled similarly to the format used in Figure S3 since the experimental TM-score differences are quite different between the different proteins. Aside from ASCT2, and potentially MCT1 with templates, the dynamic range of the conformational change appears to be minimal, but this may just be difficult to see due to the current plot format. In addition, move Figure S1 into this main figure to allow the reader to discern the structural and conformational variability in this test set. Finally, please add all of the pdbs used for the experimental comparison structures, both in this figure, and in the methods.

(1.2) We have modified Figure 1 in accordance with these recommendations (please see 0.1.5 above).

3. Is the term “ground truth structures” referring to the crystal structures or other experimental structures? Please change this term as experimental structures do not correspond to a “truth” but is a physically accessible conformational state of the protein under those experimental conditions.

(1.3) We have changed the text to avoid using this term in the text:

“For most proteins considered, we report a striking correlation between the breadth of structures predicted by AF2 and the cryo-EM and/or X-ray crystal structures.”

“To quantitatively place the predicted conformational variance in the context of the experimentally determined structures, we used principal component analysis (PCA), which reduces the multidimensional space to a smaller space representative of the main conformational motions.”

“(In Figure 3B) – Scatter plots comparing each model's position along PC1 and its structural similarity to experimentally determined structures.”

Reviewer #2 (Recommendations for the authors):1. First, I would disagree with the title of Figure S5 and the beginning of the corresponding title, which seem to be categorical about the lack of exploration of alternate conformations for these examples, but then somewhat contradict the rest of the paragraph, where it is explained that some cases (MurJ and CCR5) behave differently from the others. These discrepancies should be resolved.

(2.1) We have changed the title of the section per the Reviewer’s comments:

“Limited conformational sampling is observed for proteins with structures in the training set”

We have also rewritten the paragraph to better reflect the extent to which models generated with shallow MSAs sample alternative conformations:

“The conformational diversity of these models, including those generated using shallow MSAs, was far more limited than what was generally observed for the proteins discussed above, with the exception of MCT1 (Figure 1 —figure supplement 2). Although conformational diversity was demonstrated to a limited extent by the generation of occluded models of MurJ and PfMATE, none of the models observed adopted the alternative conformer. By contrast, while models of CCR5 were less biased toward the training set conformation, deep MSAs reduced conformational diversity. This divergence in performance may stem from the composition of the AF2 training set, which featured the structures of many active GPCRs but no structures, for example, of inward-facing MATEs^45^.”

Finally, we edited a statement in “Concluding remarks”:

“Moreover, this approach showed limited success when applied to transporters whose structures were used to train AF2, hinting at the possibility that traditional methods may still be required to capture alternative conformers^46,47^.”

2. Second, I think it would be of value for the readership to mention that no function is included to describe the membrane in these modelling processes – even when the lipids themselves may be critical to shift these conformational equilibria. This observation actually makes the authors’ findings all the more remarkable, but also perhaps harder to interpret.

(2.2) We have added a sentence to reflect the membrane’s absence from structural inference:

“Several preprints have provided evidence that AF2, despite its accuracy, likely does not learn the energy landscapes underpinning protein folding and function^35,49,50^. Moreover, AF2 does not directly account for the lipid environment, which have been experimentally shown to bias the conformational equilibria of membrane proteins^51–53^. As our results show that exploration of the conformational space is in part a byproduct of low sequence information provided for inference, they highlight the need for further development of artificial intelligence methods capable of learning the conformational flexibility intrinsic to protein structures.”

Reviewer #3 (Recommendations for the authors):1. Change Lat1 -> LAT1

(3.1.1) We have made the modification to the name of the protein throughout the text, Figures, and Tables.

2. The following statement is unclear and should be elaborated. “Several preprints have provided evidence that AF2, despite its accuracy, likely does not learn the energy landscapes underpinning protein folding and function39,53,54. We believe that our results bolster these conclusions and highlight the need for further development of artificial intelligence methods capable of learning the conformational flexibility intrinsic to protein structures.”

(3.1.2) In response to another comment made by Reviewer #2, we have edited this final paragraph (see 2.2 above).

[Editors’ note: further revisions were suggested prior to acceptance, as described below.]

The manuscript has been improved but there are some remaining issues that need to be addressed, as outlined below:1. The reviewers found that the current title may lead the reader to misinterpretation. An alternate title “Sampling alternative conformational states of transporters and receptors with AlphaFold2” is more appropriate and should be adopted.

We have changed the manuscript title as recommended.

2. The findings in Figure 1 Suppl2, that the number of sequences isn’t correlated with an increase in conformational homogeneity, and displays erratic dependence for some proteins (especially ASCT2, Lat1 and STP10), are surprising. Consequently, it seems necessary to alter some statements in the manuscript, accordingly. For example, in the abstract: “reducing the depth of the input MSAs is conducive to the generation of accurate models in multiple conformations by AF2”.

We have rewritten several statements throughout the text per the Reviewers' recommendations:

Abstract: “Whereas models of most proteins generated using the default AF2 pipeline are conformationally homogeneous and nearly identical to one another, reducing the depth of the input multiple sequence alignments (MSAs) by stochastic subsampling led to the generation of accurate models in multiple conformations.”

Introduction: “Our results demonstrate that reducing the depth of the input MSAs is often conducive to the generation of accurate models in multiple conformations by AF2, suggesting that the algorithm's outstanding predictive performance can be extended to sample alternative structures of the same target.”

Results and Discussion: “Decreasing the depth of the subsampled MSAs, by contrast, appeared to promote the generation of alternative conformations in most proteins.”

Results and Discussion: “The use of shallow MSAs was instrumental to obtaining structurally diverse models in most proteins, and in one case (MCT1) accurate modeling of alternative conformations also required the manual curation of template models.”